# Ultra-flat and long-lived plasmons in a strongly correlated oxide

Han Gao[1,7], Chao Ding[1,7], Jaeseok Son [2,3], Yangyu Zhu[1], Mingzheng Wang[1], Zhi Gen Yu [4], Jianing Chen[5], Le Wang [6], Scott A. Chambers[6], Tae Won Noh [2,3], Mingwen Zhao [1] ✉ & Yangyang Li [1] ✉

Plasmons in strongly correlated systems are attracting considerable attention due to their unconventional behavior caused by electronic correlation effects. Recently, flat plasmons with nearly dispersionless frequency-wave vector relations have drawn significant interest because of their intriguing physical origin and promising applications. However, these flat plasmons exist primarily in low-dimensional materials with limited wave vector magnitudes ($q < {\sim}0.7$ Å$^{-1}$). Here, we show that long-lived flat plasmons can propagate up to ${\sim}1.2$ Å$^{-1}$ in α-Ti$_2$O$_3$, a strongly correlated three-dimensional Mott-insulator, with an ultra-small energy fluctuation (<40 meV). The strong correlation effect renormalizes the electronic bands near Fermi level with a small bandwidth, which is responsible for the flat plasmons in α-Ti$_2$O$_3$. Moreover, these flat plasmons are not affected by Landau damping over a wide range of wave vectors ($q < {\sim}1.2$ Å$^{-1}$) due to symmetry constrains on the electron wavefunctions. Our work provides a strategy for exploring flat plasmons in strongly correlated systems, which in turn may give rise to novel plasmonic devices in which flat and long-lived plasmons are desirable.

Strongly correlated insulators are materials that are expected to be metals according to conventional band theory but are actually insulators with half-filled orbitals or unpaired electrons[1–3]. On-site Coulomb interactions ($U$) must be considered when describing the electronic band structures of correlated insulators as they are the cause of the unexpected bandgaps[3]. Based on the Mott-Hubbard model, the energy bands near Fermi level ($E_F$) are renormalized because of $U$, resulting in an upper Hubbard band (UHB) and a lower Hubbard band (LHB) (Supplementary Fig. 1)[4–6]. Depending on the difference between $U$ and the charge-transfer energy ($\Delta$), strongly correlated oxides are classified as either charge-transfer insulators ($U > \Delta$) or Mott insulators ($U < \Delta$). Several fascinating properties have been observed in strongly correlated oxides, including correlated topological phases[7,8], metal-insulator transitions (MIT)[3], and unconventional superconductivity[9],

among others. As collective motions of electrons, plasmons arise from long-range Coulomb interaction. In strongly correlated electron systems, plasmonic behavior can be drastically altered due to the strong correlation effects, leading to novel properties and unprecedented functionalities[10–12]. For example, correlation effect with long-range Coulomb interactions could induce unconventional correlated plasmons with multiple plasmon frequencies and low-loss[11,12].

As a general rule, the frequency of plasmons, $\omega_p$, strongly depends on the wave vector $q$. In traditional three-dimensional (3D) metals, the relationship is $\omega_p = \omega_0 + (3v_F^2/10\omega_0)q^2$ in the long-wavelength limit[13], where $\omega_0$ is the plasmon frequency for $q \to 0$, and $v_F$ is the Fermi velocity. In a two-dimensional (2D) system such as graphene, it follows that $\omega_p \propto \sqrt{q}$ when $q \to 0$[14], as in an ideal 2D electron gas. Similar plasmon behavior ($\omega_p \propto \sqrt{q}$) has been reported in

[1]School of Physics, Shandong University, Jinan 250100 Shandong, China. [2]Center for Correlated Electron Systems, Institute for Basic Science (IBS), Seoul 08826, Republic of Korea. [3]Department of Physics and Astronomy, Seoul National University, Seoul 08826, Republic of Korea. [4]Institute of High Performance Computing, Singapore 138632, Singapore. [5]Institute of Physics, Chinese Academy of Sciences and Collaborative Innovation Center of Quantum Matter, Beijing 100190, China. [6]Physical and Computational Sciences Directorate, Pacific Northwest National Laboratory, Richland, WA 99354, USA. [7]These authors contributed equally: Han Gao, Chao Ding. ✉e-mail: zmw@sdu.edu.cn; yangyang.li@sdu.edu.cn

2D metallic monolayers and quasi-2D metals at small $q$. However, the plasmons become dispersionless (referred hereafter as flat plasmons) over a relatively large range of wave vectors (0.1 Å$^{-1}$–0.3 Å$^{-1}$)[15–17] due to the screening effects arising from the interband transitions[18–20]. Additionally, flat plasmons have been reported in twisted bilayer graphene[21–23]. Interestingly, flat plasmons can transition to localized plasmon wave packets in real-space. By tracking these plasmon wave packets, novel time-resolved plasmonic imaging technique could be realized[18]. However, flat plasmons have been reported to date only over a limited range of wave vectors, $q$ <- 0.7 Å$^{-1}$ [15–17], and exclusively in 2D or quasi-2D systems[18,21,22].

In this work, we report on flat plasmons that can propagate up to an ultra-large wave vector, $q$ > 1.2 Å$^{-1}$ (beyond the first Brillouin zone), with a small energy fluctuation of less than 40 meV in a strongly correlated 3D oxide, α-Ti$_2$O$_3$. α-Ti$_2$O$_3$ is a typical Mott-insulator with strongly correlated 3$d^1$ electrons, exhibiting a broad MIT above 400 K[24–26]. As a consequence, α-Ti$_2$O$_3$ has a narrow bandgap of ~0.1 eV at room temperature which in turn gives rise to fascinating physical properties and applications[27–29], such as high-performance mid-infrared photodetection[27] and photothermal conversion[28,29]. Moreover, novel superconductivity[30] and interesting catalytic properties[31] have been reported in newly epitaxial stabilized Ti$_2$O$_3$ polymorphs, and these are closely related to the electronic correlations therein. Because of strong electronic correlation effects, the energy bands near $E_F$ are renormalized and become relatively flat, leading to flat plasmons in α-Ti$_2$O$_3$. Due to the crystal symmetry and negligible absorption at the plasmonic frequency, these plasmons can propagate beyond the first Brillouin zone. Additionally, we present evidence for a hyperbolic property arising from the anisotropic electronic structures of α-Ti$_2$O$_3$.

## Results

α-Ti$_2$O$_3$ has a corundum structure with the space group $R\bar{3}c$. The conventional and primitive cells of Ti$_2$O$_3$ are shown in Fig. 1a, b, respectively. The conventional cell has a hexagonal representation with lattice parameters $a$ = $b$ = 5.15 Å and $c$ = 13.64 Å[32]. The primitive cell is a rhombohedra with $a_1$ = $a_2$ = $a_3$ = 5.517 Å and an angle of 55.2° between lattice vectors. Each unit has a titanium atom surrounded by six oxygen atoms, constituting a distorted octahedral configuration. The four titanium atoms in the primitive cell lie adjacent along the $c$-axis, forming Ti-Ti dimers with face-shared octahedra[33]. Figure 1c shows high-resolution X-ray diffraction (HR-XRD) scans for the α-Ti$_2$O$_3$ single crystal with (0006) and (11$\bar{2}$0)-oriented surface planes; these

scans confirm the hexagonal characteristics of α-Ti$_2$O$_3$. (More structural details are provided in Supplementary Fig. 2).

Within the hexagonal structure, there are face-sharing octahedra along the $c$-axis and edge-sharing octahedra in the $ab$-plane. Strong $d$-$d$ orbital interaction arises from the Ti atoms with face-sharing and edge-sharing octahedra due to the short Ti-Ti distances which in turn form bonding and antibonding molecular orbitals[24,33]. The proposed molecular orbital diagram for α-Ti$_2$O$_3$, based on Goodenough's model, is shown as an inset in Fig. 1d. The states near $E_F$ are dominated by Ti 3$d$ orbitals whereas the O 2$p$ orbitals are far below $E_F$, consistent with our theoretical calculations (Supplementary Fig. 3). The bonding $a_{1g}$ and antibonding $a_{1g}^*$ molecular orbitals are directed along the $c$-axis with the bonding $e_g^{\pi}$ and antibonding $e_g^{\pi*}$ molecular orbitals lying in the $ab$-plane, leading to an anisotropic band structure near $E_F$[25,33]. The $a_{1g}$ band is fully occupied with Ti 3$d^1$ electrons and is well separated from the $a_{1g}^*$ band. (The splitting between the $e_g^{\pi}$ and $e_g^{\pi*}$ bands is negligible[33]). Due to the $U$ value for the 3$d^1$ electrons, the $a_{1g}$ band is separated from the $e_g^{\pi}$ band, resulting in a gap of ~0.1 eV at room temperature. Figure 1d shows the temperature-dependent resistivity for α-Ti$_2$O$_3$ with $I || xy$-plane and $I || z$. Clearly, its resistivity is increased with decreasing temperature, which attests to the correlated insulating behavior of α-Ti$_2$O$_3$[24–26]. Moreover, the anisotropic behavior for the two directions of current flow is consistent with the anisotropic band structure near $E_F$.

In order to further study the anisotropic behavior of α-Ti$_2$O$_3$, we investigated its optical properties by theoretical calculations and experimental measurements. The optical properties of crystalline solids can be described by a complex permittivity $\varepsilon(\omega)_{\alpha\beta}$ = Re[$\varepsilon(\omega)_{\alpha\beta}$] + Im[$\varepsilon(\omega)_{\alpha\beta}$], where $\alpha$, $\beta$ represent the different Cartesian directions. The permittivity is the sum of the interband and intraband transition contributions. For the interband contributed part, the imaginary term Im[$\varepsilon(\omega)_{\alpha\beta}^{\text{inter}}$] can be calculated from the interband transitions, while the real term Re[$\varepsilon(\omega)_{\alpha\beta}^{\text{inter}}$] is determined from Im[$\varepsilon(\omega)_{\alpha\beta}^{\text{inter}}$] according to the Kramers-Kronig relation[34]. For the intraband contributed part, the Drude model is used to describe transitions within the partially occupied bands of the material[35],

$$\varepsilon(\omega)_{\alpha\beta}^{\text{intra}} = 1 - \frac{\omega_{p,\alpha\beta}^2}{\omega^2 + i\gamma\omega} \tag{1}$$

here, $\omega_p$ is the plasma frequency and $\gamma$ is the lifetime broadening which is the reciprocal of the excited-state lifetime.

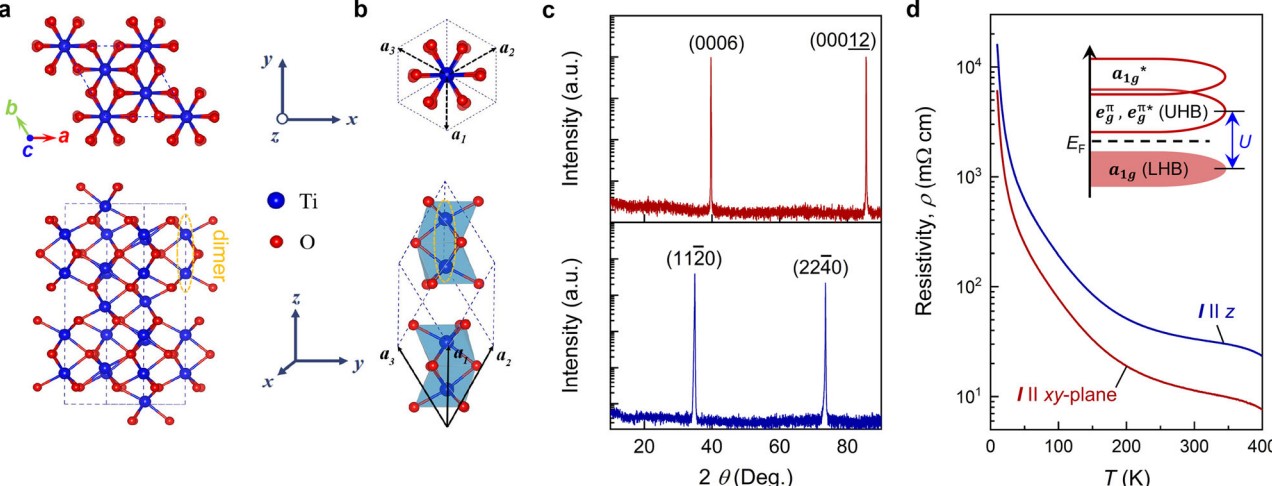

**Fig. 1 | Crystal structure and correlated insulating behavior for α-Ti$_2$O$_3$.** Schematic views of the **a** conventional cell and **b** unit cell of α-Ti$_2$O$_3$. The upper panel is a top view and the lower panel is a lateral view. Ti and O atoms are represented by the blue and red balls, respectively. **c** HR-XRD patterns for a α-Ti$_2$O$_3$ single crystal with (0001) and (11$\bar{2}$0)-oriented surfaces. **d** Temperature-dependent resistivity for a α-Ti$_2$O$_3$ single crystal, collected with $I || xy$-plane and $I || z$ orientations, where $I$ is the applied electrical current.

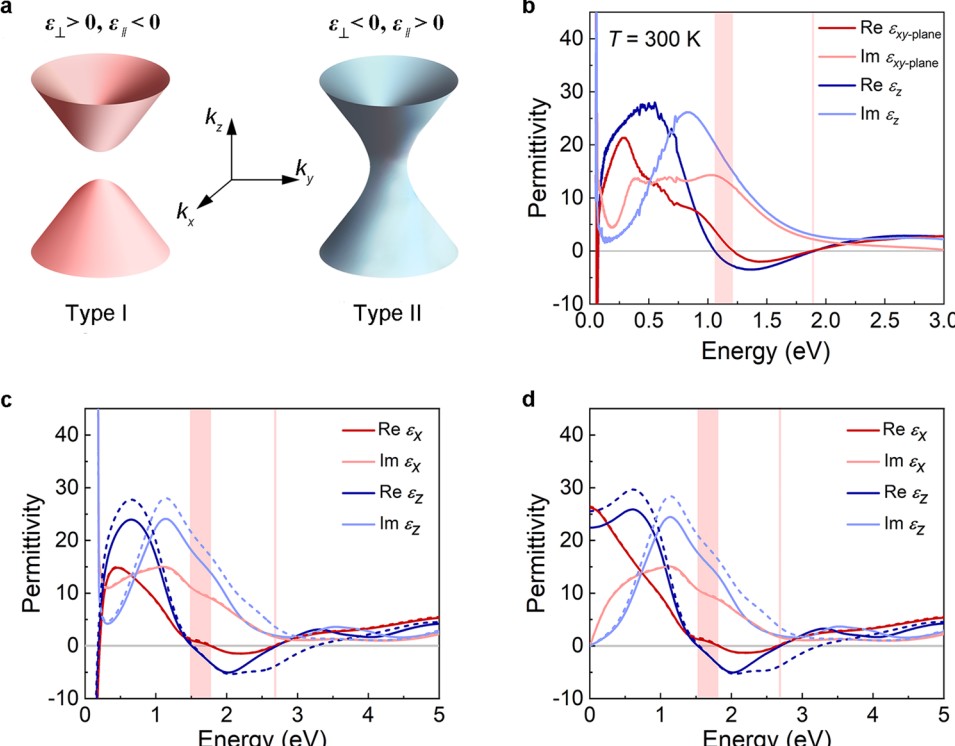

**Fig. 2 | Dielectric properties of α-Ti₂O₃ crystal. a** Schematic diagrams of type-I and type-II hyperbolic materials. **b** Real and imaginary parts of the permittivity along the *xy*-plane and the *z*- direction, collected by ellipsometry at room temperature. **c** Real and imaginary parts of the permittivity of α-Ti₂O₃ obtained from first-principles calculations. **d** Real and imaginary parts of the permittivity

contributed exclusively by interband transitions. The pink shaded regions show the hyperbolic frequency windows, where the real parts of $\varepsilon_\parallel$ and $\varepsilon_\perp$ have opposite signs. The data indicated by the solid (dashed) lines are results with (without) LFE. Source data are provided as a Source Data file.

For a non-magnetic bulk material which is anisotropic along the in-plane and out-of-plane directions, the isofrequency surface for transverse magnetic (TM) polarized waves ($k_x$, $k_y$, $k_z$) is given by

$$\frac{k_x^2 + k_y^2}{\varepsilon_\parallel} + \frac{k_z^2}{\varepsilon_\perp} = \left(\frac{\omega}{c}\right)^2 \tag{2}$$

where $c$ is the speed of light. $\varepsilon_\parallel$ and $\varepsilon_\perp$ denote the components of permittivity tensor parallel and perpendicular to the anisotropy axis, respectively. If the real parts of $\varepsilon_\parallel$ and $\varepsilon_\perp$ have opposite signs in a medium, the isofrequency surface will be a hyperboloid and this class of materials is known as "hyperbolic materials" (HMs)[36]. Furthermore, HMs can be classified into type I ($\varepsilon_\perp > 0$ and $\varepsilon_\parallel < 0$) and type II ($\varepsilon_\perp < 0$ and $\varepsilon_\parallel > 0$) HMs[37]. Schematic diagrams of the dispersion relations for these two types of HMs are shown in Fig. 2a. Type I HMs usually have fewer reflections and possess lower losses than do type II HMs[38].

For α-Ti₂O₃, the permittivity is isotropic in the *xy*-plane, and anisotropic along the *z* direction (*c*-axis) (i.e. $\varepsilon_\perp = \varepsilon_x$ and $\varepsilon_\parallel = \varepsilon_z$). Figure 2b presents the real part and imaginary part of permittivity for α-Ti₂O₃ measured by ellipsometry at room temperature with **E**||*xy*-plane and **E**||*z*. Two type-I hyperbolic regions at 1.06−1.21 eV and 1.88−1.90 eV with Re$\varepsilon_x > 0$ and Re$\varepsilon_z < 0$ are observed experimentally. Moreover, both Im$\varepsilon_x$ and Im$\varepsilon_z$ in the second hyperbolic region present small values where Re($\varepsilon$) changes sign, which is well consistent with the theoretical results (Fig. 2c, d). The only differences are the absolute frequency values. In Fig. 2c, without considering the local field effect (LFE) (dashed lines), there are two hyperbolic windows at the energy range of 0−5 eV. In the first hyperbolic window at 1.52−1.79 eV, the imaginary part of permittivity Im$\varepsilon(\omega)$ has a relatively large value, indicating a large energy loss due to the electron transitions. But in the second hyperbolic region of 2.67−3.26 eV, Im$\varepsilon(\omega)$ is greatly

suppressed, making α-Ti₂O₃ an ideal type I hyperbolic material within the corresponding photon energy range. When considering the LFE (solid lines), consistent with the experimental results, the second hyperbolic window shrinks to almost disappear because of the redshifted frequency of Re$\varepsilon_z(\omega) = 0$ from 3.26 eV to 2.69 eV. The large correction effect of LFE on $\varepsilon_z(\omega)$ can be attributed to the unhomogeneous distribution of the wave function along the *z* direction[33], which causes the off-diagonal terms of $\varepsilon_{G=G'}(q,\omega)$.

Multiple Re$\varepsilon = 0$ points are observed in the experimental[39] and theoretical results[40], and this unusual predominance making them applicable as functional hyperbolic metamaterials[39]. Figures 3a, b present the plasmon properties of α-Ti₂O₃ in the energy range of 0−5 eV, obtained from the calculated electron energy loss spectra (EELS) along the in-plane (Γ−$S_O$) and out-of-plane (Γ−$T$) directions. The plasmon dispersion is shown, as extracted from the peak values of the EELS. Surprisingly, the plasmons along the in-plane and out-of-plane directions are both nearly dispersionless, with a small energy fluctuation $\Delta\omega_p < 40$ meV. For the Γ−$S_O$ direction (Fig. 3a), the plasmon mode starts at ~2.60 eV, close to the frequency where Re$\varepsilon_x(\omega) = 0$ (Fig. 2c). Importantly, the plasmons can propagate over a large momentum range and remain visible well beyond the first Brillouin zone ($q < $ ~1.2 Å⁻¹). Notably, the nearly dispersionless behavior can be observed from the excitation spectrum and the maximum change of the plasmon energy is only $\Delta\omega_p(Γ−S_O) = 40$ meV within the first Brillouin zone. To verify the collective excitation and long-lived features of the plasmon modes, we plot the dielectric function and loss function at selected momenta, $q = 0.077$ Å⁻¹ (Fig. 3c) and $q = 0.77$ Å⁻¹ (Fig. 3d). For an undamped plasmon, the dielectric function should fulfill the condition Re$\varepsilon = 0$ with $Im\varepsilon/\partial_\omega Re\varepsilon > 0$ at the peak energy in the loss function[41,42]. At the same time, $Im\varepsilon$ should have a vanishing value, indicating the plasmon is free of Landau damping[42]. For $q = 0.077$ Å⁻¹ along the Γ−$S_O$ direction (Fig. 3c), Re$\varepsilon$ crosses zero from negative to

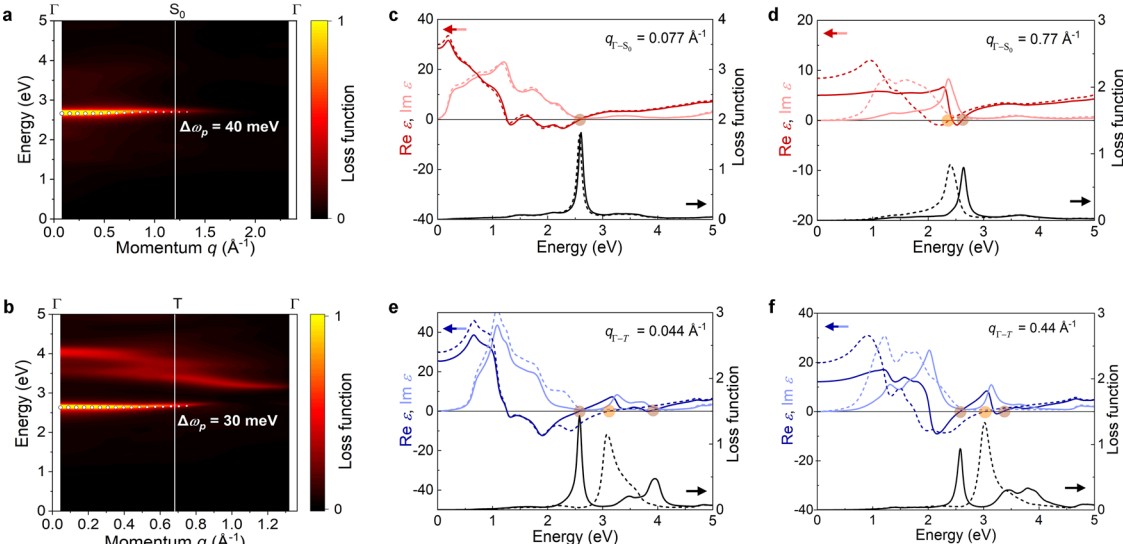

**Fig. 3 | Plasmon behavior for α-Ti₂O₃. a, b** Calculated EELS with LFE for α-Ti₂O₃ along Γ-$S_O$ and Γ-$T$ directions, respectively. White circles outline the plasmon branches extracted from EELS and the size represents the intensity of plasmon modes. **c, d** Dielectric function (upper panel) and loss function (lower panel) of α-Ti₂O₃ for $q = 0.077$ Å⁻¹ and $q = 0.77$ Å⁻¹ along the Γ-$S_O$ direction, respectively.

**e, f** Dielectric function (upper panel) and loss function (lower panel) of α-Ti₂O₃ for $q = 0.044$ Å⁻¹ and $q = 0.44$ Å⁻¹ along the Γ-$T$ direction, respectively. The data indicated by the solid (dashed) lines are results with (without) the LFE. The circles denote the zeros of Re$\varepsilon$($\mathbf{q}$,$\omega$). Source data are provided as a Source Data file.

positive values at $\omega_p = $ ~2.60 eV with a vanishing Im$\varepsilon$, which corresponds to the energy of the sharp peak in the loss function. For $q = 0.77$ Å⁻¹ (Fig. 3d), those conditions are also satisfied, verifying the robustness of the undamped plasmon along the Γ-$S_O$ direction. Additionally, the energy of the in-plane plasmon (Γ-$S_O$) is blue shifted from 2.37 eV to 2.61 eV (solid lines in Fig. 3c, d) at large $q = 0.77$ Å⁻¹, while that at small $q = 0.077$ Å⁻¹ is negligible, as the LFE is considered.

For the Γ−$T$ direction (Fig. 3b), the plasmon exhibits some behavior similar to that along the Γ-$S_O$ direction. The out-of-plane plasmon propagates from ~2.58 eV without dispersion and also persists beyond the first BZ. In contrast to the EELS along the Γ−$S_O$ direction, there are some additional broad and weak peaks at higher energies in the excitation spectrum (Fig. 3b). Nevertheless, these peaks are damped modes, which originate from single particle excitations[41]. According to the dielectric function and loss function at selected momenta (Fig. 3e, f), the LFE has large effects on the out-of-plane plasmon mode. Without the LFE, there is only one peak in the loss function, located at 3.08 eV and 3.02 eV for $q = 0.044$ Å⁻¹ and $q = 0.44$ Å⁻¹, respectively. When the LFE is taken into account, however, the peak splits into two features, a sharper one at a lower energy of 2.58 eV (2.59 eV) and a broader one at higher energy at 3.95 eV (3.46 eV) for $q = 0.044$ Å⁻¹ ($q = 0.44$ Å⁻¹). The lower energy mode is stronger and meets the conditions for the undamped plasmons, contributing to the long-lived and well-defined out-of-plane plasmon (Fig. 3b). For the higher energy mode, although the definition of the plasmon Re$\varepsilon = 0$ is satisfied, the continuum character of Im$\varepsilon$ and the small derivative of Re$\varepsilon$ in the nearby region indicates the rapid decay of this mode into electron-hole pairs[42].

It is noteworthy that the ultra-flat dispersion of plasmons is rare in most materials, and this fact motivates us to further study its physical origins in this 3D oxide. Since the ultra-flat plasmon comes mainly from the absorption peak centered at ~1.2 eV in the $Im\varepsilon(\omega, q \rightarrow 0)$ (Fig. 2c), we first analyze the origin of the absorption peak from the electronic transition processes. The orbital-resolved electronic band structure of α-Ti₂O₃ is plotted in Fig. 4a. The two bands below $E_F$ are largely derived from the Ti $3d_{z^2}$ orbital, whereas the conduction bands near $E_F$ originate mainly from the Ti$3d_{xy}$ and $3d_{x^2-y^2}$ orbitals. We analyzed the symmetry of the wave functions at the Γ point using the Irvsp[43] code. The point group of α-Ti₂O₃ is $D_{3d}$, which contains the

space inversion operation $P$ respect to $O$ point (Fig. 4b). The degenerate states at Γ can be used as the basis functions for the construction of the irreducible representations of the $D_{3d}$ point group. The corresponding irreducible representations for states 1-12 at the Γ point are listed in Table 1. Due to the space inversion symmetry of α-Ti₂O₃, the wave functions for Γ point have certain parities of space inversion operation $P$. The parities of the wave functions are also presented, and these are the eigenvalues of the space inversion operation $P$. The transition dipole moment matrix associated with the transition between an initial states $m$ and $n$ is defined as $\langle\psi_{m\Gamma}|\mathbf{r}|\psi_{n\Gamma}\rangle$, where $\psi_{m\Gamma}$ and $\psi_{n\Gamma}$ are the electron wave functions, and $\mathbf{r}$ is the position operator. For α-Ti₂O₃, it shows a space inversion symmetry with respect to coordinate $O$ point (Fig. 4b). If $\psi_{m\Gamma}$ has the same parity as $\psi_{n\Gamma}$ with respect to $O$ point, we will have $\langle\psi_{m\Gamma}|\mathbf{r}|\psi_{n\Gamma}\rangle = 0$. Thus, only the interband transitions between the two states with opposite parities are allowed and have contributions to the Im$\varepsilon$. In Table 1, we summarize the allowed and forbidden electric-dipole transitions at the Γ point according to the optical selection rule. We further calculated the matrix element $\langle\psi_{m\Gamma}|\mathbf{r}|\psi_{n\Gamma}\rangle$ for transitions between different states at the Γ point. According to our calculations, the transitions from state 2 below $E_F$ to the degenerate states 3 and 4 exhibit the largest matrix elements, whereas the matrix elements of other transitions are almost negligible (Supplementary Table 1). It should be mentioned that the flat plasmons in α-Ti₂O₃ along both directions are located at ~2.6 eV. Thus, its Landau damping should be arisen from the interband transitions with the transition energy close to 2.6 eV. As shown in Fig. 4a, only the transition between state 1 and state 12 has the corresponding energy difference. However, the wave functions of state 1 and state 12 both exhibit an even parity, so this transition is forbidden by the selection rule (Table 1). Hence, we get vanished values for Im($\varepsilon$) at the plasmon energy, which preserves the plasmons are long-lived. Finite-temperature effect on the damping of the plasmons in α-Ti₂O₃ is discussed in the supplementary materials (Supplementary Fig. 4–6, Supplementary Note 1).

In the long-wavelength limit, the imaginary part of the interband dielectric function is proportional to the joint density of states (JDOS) and the transition matrix elements[44]. The JDOS is defined as $D_{JDOS}(E) = \frac{1}{(2\pi)^3}\sum_{c,v}\int_E \frac{dS_k}{|\nabla_k(E_{k,c}-E_{k,v})|}$, where $E_{k,c}$ and $E_{k,v}$ are energies in

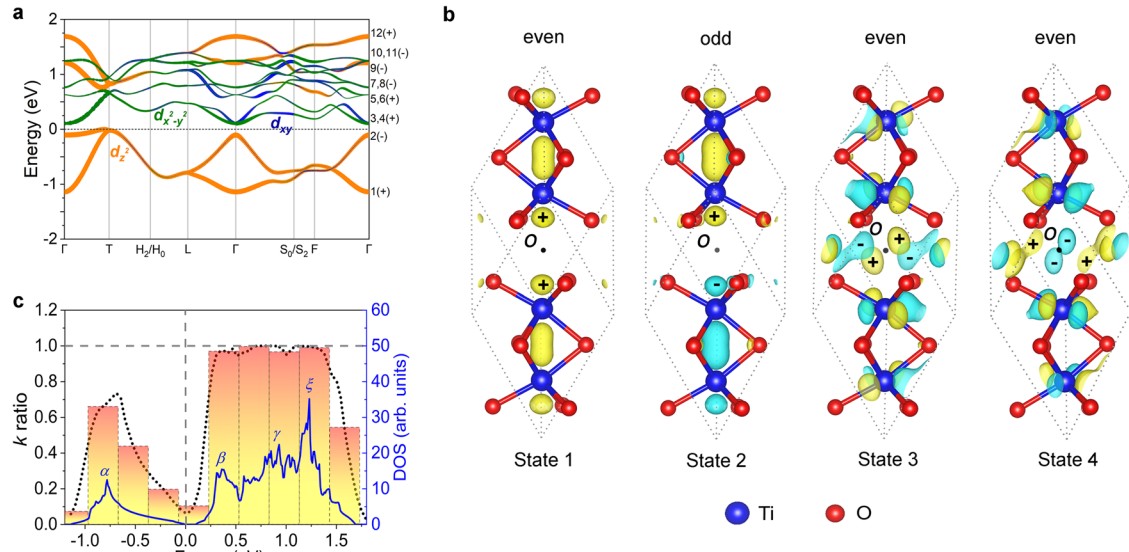

**Fig. 4 | Origin of the ultra-flat plasmon in α-Ti₂O₃.** **a** Orbital-resolved band structure of α-Ti₂O₃. The contributions from the Ti $3d_{z^2}$, $d_{xy}$ and $3d_{x^2-y^2}$ orbitals are denoted by orange, blue and green curves, respectively. **b** The Bloch electron wave functions corresponding to the state 1, 2, 3, 4 at the Γ point. $O$ point is the center of inversion symmetry. Yellow (+) and blue (−) color represent the sign of the real part of the wave functions, respectively. **c** The density of states (DOS) (blue solid line) and $k$ ratio (black dotted line) as a function of energy. The height of the histogram is determined by the $k$ ratio for energy at the center of the histogram. Source data are provided as a Source Data file.

the conduction and valence band, respectively, and $S_k$ is the constant-energy surface defined by $E_{k,c} − E_{k,v} = E$. In Fig. 4c, there is one peak $\alpha$ below $E_F$ and three main peaks ($\beta$, $\gamma$ and $\xi$) above $E_F$ in the energy range −2 eV to 2 eV. Obviously, $\alpha \to \beta$, $\alpha \to \gamma$ and $\alpha \to \xi$ are the three possible transitions that can contribute significantly to the JDOS. Among the three transitions, only $\alpha \to \beta$ has a transition energy close to 1.2 eV. Based on the electronic band structure (Fig. 4a) and the parities of the Bloch electron wave functions (Table 1), we attribute the absorption peak to the transitions between state 2 and states 3 & 4, corresponding to the transition from $a_{1g}$ to $e_g^\pi$ (inset of Fig. 1d)[25].

Having discussed the electronic transitions corresponding to the plasmons, we now consider the origin for their ultra-flat behavior. In general, the plasmon modes can be obtained by solving $\det|\varepsilon_{G,G'}(q,\omega)| = 0$, with the dielectric function matrix element $\varepsilon_{GG'}(q,\omega) = \delta_{G,G'} − v(q+G)\chi^0_{G,G'}(q,\omega)$[45,46]. The flat plasmon is attributed to a weak dependence of the dielectric function $\varepsilon_{GG'}(q,\omega)$ on $|q| = q$ along a particular direction. Although LFE shows large effect on the plasmon behavior along the out-of-plane direction (Fig. 3e, f). However, from $q_{\Gamma\text{-}T} = 0.044$ Å⁻¹ to $q_{\Gamma\text{-}T} = 0.44$ Å⁻¹, the variation of the plasmon energy without LFE is only 62 meV (3.079 eV−3.017 eV), which

indicates the plasmon mode already exhibits a dispersionless behavior without considering LFE. Hence, when we study the origin of the flat behavior for the plasmons, we neglect the LFE and reduce the dielectric function matrix to $\varepsilon_{00}(q,\omega)$,

$$\varepsilon_{00}(q,\omega) = 1 − v(q)\frac{1}{V}\sum_k^{BZ}\sum_{n,n'}\frac{f_{n,k} − f_{n',k+q}}{\hbar\omega + E_{n,k} − E_{n',k+q} + i\eta}\left|\langle\psi_{n,k}|e^{-iq\cdot r}|\psi_{n',k+q}\rangle\right|^2 \quad (3)$$

here, $v(q) = 4\pi e^2/\varepsilon_r q^2$ is the Fourier component of the three-dimensional Coulomb potential. For the interband transitions in the long-wavelength limit, we get $\langle\psi_{n,k}|e^{-iq\cdot r}|\psi_{n',k+q}\rangle \approx −iq \cdot \langle\psi_{n,k}|r|\psi_{n',k}\rangle$[47], and thus $|\langle\psi_{n,k}|e^{-iq\cdot r}|\psi_{n',k+q}\rangle|^2 \sim q^2$. Considering that $v(q) \sim q^{-2}$, we thus expect that the dependence of the dielectric function $\varepsilon_{00}(q,\omega)$ on $q$ to be dominated by the dependence of $E_{n',k+q}$ on $q$. Therefore, the flat behaviors of the plasmon can be attributed to the relatively flat band $E_{n,k}$. That is, the dielectric function $\varepsilon_{00}(q,\omega)$ is independent of $q$, as long as the band $E_{n,k}$ is flat, which leads to the flat plasmons. Notably, the flat plasmon can also exist when a series of flat bands coexists in an energy window, due to the sum over the band index $n'$ in Eq. (3). To explore the flatness of the bands, we plotted the $k$-ratio in Fig. 4c, which is defined as follows. For a certain energy $E$, there are a series of electronic states $\psi_{n,k}$ in the energy window ranging from $E − \Delta E$ to $E + \Delta E$. We define the number of the wave vectors $k$ corresponds to these electronic states $\psi_{n,k}$ as $N_1$, and the number of the wave vector over the entire Brillouin zone as $N$. The $k$-ratio is defined by the ratio of $N_1$ and $N$. According to this definition, if the $k$-ratio at the energy $E$ is equal to 1, there is at least one band in the energy window from $E − \Delta E$ to $E + \Delta E$ in the entire first Brillouin zone (not restricted to the highly-symmetric directions). For relatively small values of $\Delta E$, $k$-ratio = 1 indicates that the band at energy $E$ is relatively flat. In this work, we choose $\Delta E = 0.3$ eV, because the minimum bandwidth is nearly 0.3 eV. In Fig. 4c, the $k$-ratio of peaks $\beta$, $\gamma$ and $\xi$ are all very close to 1, indicating the existence of flat bands in the energy window from -0.3 to 1.3 eV above $E_F$ in α-Ti₂O₃. This result is consistent with the proposed narrow bands in α-Ti₂O₃, which are renormalized by the Hubbard $U$[48,49]. Thus, the electronic transitions from the valence bands to the nearly flat

**Table 1 | The electronic states at Γ, and their corresponding irreducible representations of D₃d, parity of space inversion**

| Band index | Reps | Parity | Transition (from 1) | Transition (from 2) |
|---|---|---|---|---|
| 1 | $A_{1g}$ | + | -- | -- |
| 2 | $A_{1u}$ | − | -- | -- |
| 3, 4 | $E_g$ | + | ✓ | ✓(main) |
| 5, 6 | $E_g$ | + | ✓ | ✓ |
| 7, 8 | $E_u$ | − | ✓ | ✓ |
| 9 | $A_{2u}$ | − | ✓ | ✓ |
| 10, 11 | $E_u$ | − | ✓ | ✓ |
| 12 | $A_{2g}$ | + | ✓ | ✓ |

The allowable (represented by ✓) and forbidden (represented by -) transitions from state 1 or 2 to the unoccupied states are also shown.

conduction bands lead to the flat behaviors of the plasmon. Moreover, the LFE can further reduce the plasmons dispersion, leading to the ultra-flat plasmons in $\alpha$-Ti$_2$O$_3$ with the energy corrugation less than 40 meV. Same conclusion can be made beyond the long-wavelength limit, which is presented in the supplementary materials (Supplementary Fig. 7, Supplementary Note 2).

To further explore the origin of the nearly flat bands, the real parts of the wave functions for state 1, 2, 3, 4 at the Γ point are plotted in Fig. 4b. Obviously, state 1, 2 consists largely of Ti $3d_{z^2}$ orbital (the $a_{1g}$ bonding molecular orbitals) character. State 1 has even parity with respect to $O$ point, whereas state 2 has odd parity with respect to $O$ point. States 3, 4 derives from Ti $3d_{xy}$ and $3d_{x^2-y^2}$ orbitals (the $e_g^\pi$ bonding molecular orbitals) and has even parity, consistent with our previous discussion. Notably, for state 1, 2 at the Γ point, there is considerable overlap of orbitals centered on different atoms, whereas the orbitals from different atoms in state 3, 4 overlap less. The relatively weak interaction between the in-plane orbitals of Ti atoms further contributes to the flatness of the conduction band ($e_g^\pi$ and $e_g^{\pi*}$ bands) near $E_F$.

Finally, we compare the flat plasmon modes in $\alpha$-Ti$_2$O$_3$ with some representative materials that have dispersionless plasmon modes[18,50–57]. We limit our comparison to materials for which the maximum change of plasmon energy ($\Delta\omega_p$) is less than 0.1 eV. The start-stop momentum and the flatness of the plasmons in these materials are shown in Fig. 5. (More details are shown in Supplementary Table 2.) Clearly, both in-plane and out-of-plane plasmons in $\alpha$-Ti$_2$O$_3$ can propagate through a larger momentum space and maintain a higher degree of localization than that of the other low dimensional materials. Noteworthily, for the other materials considered in Fig. 5, they, except VSe$_2$, are not strongly correlated systems, thus the renormalized flat energy bands do not exist in these systems. VSe$_2$ is also a $3d^1$ electron system, same as $\alpha$-Ti$_2$O$_3$. Correlation effect should also exist among those V $3d^1$ electrons, thus renormalized energy bands can be expected in VSe$_2$. And there are indeed some relatively flat unoccupied V $3d$ bands near $E_F$[57]. However, its correlation is not so strong to open a band gap at $E_F$, which makes it a metallic system. Thus, its flat plasmon originates from the intraband transition with the screening effect of the interband transitions[18,57], which is similar to those in the other non-correlated systems

but different from that in $\alpha$-Ti$_2$O$_3$. As we discussed above, $\alpha$-Ti$_2$O$_3$ is a strongly correlated insulator, and its long-lived flat plasmons originate from the interband transitions between relatively flat occupied and unoccupied bands.

Based on our analysis, at least one flat occupied band and one flat unoccupied band are needed to generate flat plasmons in $\alpha$-Ti$_2$O$_3$. Adding more flat bands near $E_F$, more absorption can be achieved, which would further change the plasmon energy and intensity. As for tuning the plasmon behaviors in $\alpha$-Ti$_2$O$_3$, we believe chemical doping (i.e. V-doping) could be an efficient way. In strongly correlated systems, chemical doping can tune the strength of correlation effect, which further tunes the band renormalization that leads to varied bandwidth, band position, band gap, and thus absorption[25]. Since the flat plasmons in $\alpha$-Ti$_2$O$_3$ originate from the interband transitions between those correlation-effect renormalized occupied and unoccupied flat bands, their behaviors (including plasmon energy, flatness and intensity) could be tuned by chemical doping.

## Discussion

In summary, we systematically investigated the electronic structure and plasmonic properties of the strongly correlated 3D oxide $\alpha$-Ti$_2$O$_3$. Our results show that $\alpha$-Ti$_2$O$_3$ possesses long-lived plasmons in both in-plane and out-of-plane directions with propagation momentum over the first Brillouin zone. Moreover, plasmon modes in $\alpha$-Ti$_2$O$_3$ exhibit ultra-flat behavior, with an energy fluctuation of less than 40 meV. We correlate these intriguing plasmons in $\alpha$-Ti$_2$O$_3$ to the relatively flat conductive bands that are renormalized by the strong electron-electron interaction $U$ and present a general physical mechanism for this effect. Significantly, the ultra-flat feature in the plasmon dispersion in $\alpha$-Ti$_2$O$_3$ is superior to that of other low dimensional materials; the resulting highly localized and low-velocity plasmon wave packets can have considerable potential for fine electronic structure detection and electric field enhancement[18,58–60]. Notably, the mechanism presented here is universal as long as the plasmon originates from interband transitions from a relatively flat occupied band to a flat unoccupied band. Our work extends the study of flat plasmons in 3D systems and highlights the interplay of correlation effects, electronic bandwidth and plasmon dispersion in strongly correlated systems. This study will stimulate searches for and investigations of flat plasmons in other correlated systems.

## Methods

### Sample preparation and experimental characterizations

The $\alpha$-Ti$_2$O$_3$ single crystals were synthesized by mixing high-purity TiO$_2$ and TiH$_4$, and then calcining the mixture at 1000 °C in vacuum[61,62]. Prior to structural characterization and electrical and optical measurements, the single crystals were cut parallel to the (0001) and (11$\bar{2}$0) surfaces and then polished. For the XRD measurements, the sample was characterized using a Bruker D8 DISCOVER high-resolution diffractometer, which is equipped with Cu Kα radiation source and LynxEye detector. The X-ray source is operated at 40 kV and 60 mA. The resistivity vs, temperature data were taken using the standard four-probe method in a commercial Quantum Design physical property measurement system (PPMS). A commercial spectroscopic ellipsometer (M2000DI and IR-VASE Mark II; J.A. Woollam Co.) was used to measure the optical response in the $xy$-plane and along the $z$-axis of the $\alpha$-Ti$_2$O$_3$ single crystals. The measurement was operated in an ultra-high vacuum cryostat at room temperature.

### Theoretical calculations

Our first-principles calculations were performed using density functional theory, as implemented in the Vienna ab simulation package (VASP)[63] and GPAW codes[64], both of which employ the projected

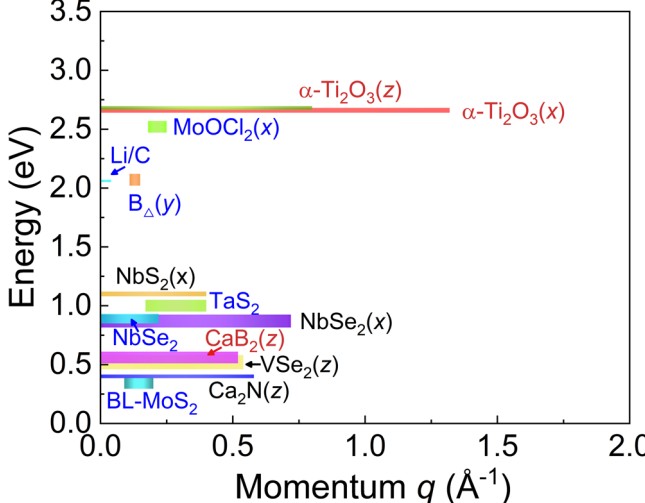

**Fig. 5 | The superiority of the plasmons in $\alpha$-Ti$_2$O$_3$.** Comparison with other materials[18,50–57] that have a relatively flat plasmon modes ($\Delta\omega_p < 0.1$ eV). The length of the bar represents the start/end momentum of the flat plasmons. The width of the bar represents the flatness of the plasmon mode $\Delta\omega_p$. Words using red, blue and black colors represent the three-dimensional, two-dimensional (monolayer or double layers) and bulk layered materials, respectively.

augmented-wave method to model interactions between electrons and ions[65]. The exchange-correlation functional was treated self-consistently within the generalized gradient approximation (GGA) using the Perdew-Burke-Ernzerhof (PBE) functional[66]. The cutoff energy was set to 500 eV. The GGA + U method[67] accounting for strong Coulomb interaction between the partially filled $3d$-shells of Ti was also employed. The Hubbard interaction parameter $U_{eff}$ (U - J, where J = 0) was set to 3.0 eV, to bring the calculated band gap closer to the experimental value[68,69]. Structure relaxation and electronic properties of $Ti_2O_3$ were calculated using VASP with the $8 \times 8 \times 8$ $(11 \times 11 \times 4)$ k-point mesh for primitive cell (conventional cell). The lattice constants and the atomic positions were fully relaxed until the atomic forces on the atoms were less than 0.01 eV/Å and the total energy change was less than $10^{-5}$ eV.

Calculations of the dynamic dielectric function and loss function were performed using linear response theory[70] implemented in the GPAW code. The conventional cell for $Ti_2O_3$ was used to calculate the $q \to 0$ limited dielectric function for different directions along the principal axis. A denser $k$ mesh of $32 \times 32 \times 10$ was adopted to converge the optical calculations. In order to conserve computing resources, the primitive cell was used in calculating the $q$-dependent loss function. Two orthogonal directions along $\Gamma - S_O$ and $\Gamma - T$ were chosen with a dense $k$-point grid of $31 \times 31 \times 31$. Under the random phase approximation (RPA), the dielectric matrix for wave vector $\mathbf{q}$ was represented as:

$$\varepsilon_{G,G'}^{RPA}(\mathbf{q},\omega) = \delta_{G,G'} - \frac{4\pi}{|\mathbf{q}+\mathbf{G}|^2}\chi_{G,G'}^0(\mathbf{q},\omega)$$

where $\chi_{G,G'}^0$ is the non-interacting density response function in reciprocal space, written as[45,46]

$$\chi_{GG'}^0(\mathbf{q},\omega) = \frac{1}{\Omega}\sum_{\mathbf{k}}^{BZ}\sum_{n,n'}\frac{f_{n\mathbf{k}}-f_{n'\mathbf{k}+\mathbf{q}}}{\omega+\varepsilon_{n\mathbf{k}}-\varepsilon_{n'\mathbf{k}+\mathbf{q}}+i\eta} \times \left\langle\psi_{n\mathbf{k}}|e^{-i(\mathbf{q}+\mathbf{G})\cdot\mathbf{r}}|\psi_{n'\mathbf{k}+\mathbf{q}}\right\rangle\Omega_{cell}$$

$$\times \left\langle\psi_{n\mathbf{k}}|e^{i(\mathbf{q}+\mathbf{G}')\cdot\mathbf{r}'}|\psi_{n'\mathbf{k}+\mathbf{q}}\right\rangle\Omega_{cell}$$

where $\mathbf{G}$ and $\mathbf{q}$ are the reciprocal lattice vector and wave vector, respectively. $f$ is the Fermi distribution function calculated by the following formula

$$f(E) = \frac{1}{1+\exp[(E-E_F)/k_BT]}$$

The Kohn-Sham energy eigenvalues $\varepsilon_{n\mathbf{k}}$, the wave function $\psi_{n\mathbf{k}}$ and the Fermi distribution function $f_{n\mathbf{k}}$ for the $n$th band at wave vector $\mathbf{k}$ were obtained from the ground-state calculations. The electron energy loss spectrum (EELS) can be calculated from the inverse of the macroscopic dielectric matrix $\varepsilon_M(\mathbf{q},\omega) = 1/\varepsilon_{G=G'=0}^{-1}(\mathbf{q},\omega)$

$$L(q,\omega) = -Im[1/\varepsilon_M(q,\omega)]$$

The plasmon energy was then extracted from local maxima in the EELS. In our calculations, 84 empty bands were considered to describe the response function. The broadening parameter $\eta$ was taken to be 0.05 eV. A cut-off of 50 eV was used to account for local field effects. The irreducible representations and parity of electronic states was computed using the Irvsp[43] code.

## Data availability
All relevant data presented in this manuscript are available from the authors upon reasonable request. The source data underlying Figs. 2b, d, 3, 4a, 4c, and Supplementary Fig. 5 are provided as a Source Data file. Source data are provided with this paper.

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

## Acknowledgements

This work was supported by the Qilu Young Scholars Program of Shandong University, the National Natural Science Foundation of China (no. 12074218), and the Taishan Scholar Program of Shandong Province. J.S.S. and T.W.N. acknowledge the support from the Research Center Program of the IBS (Institute for Basic Science) in Korea (grant no. IBS-R009-D1). Z.G.Y. acknowledge supports from the Science and Engineering Research Council of Singapore with Grant No. A1898b0043 and Grant No. 152-70-00017, and computational resource was provided by A*STAR Computational Resource Centre, Singapore (A*CRC) and the National Supercomputing Centre Singapore (NSCC). The work at PNNL was supported by the U.S. Department of Energy, Office of Science, Division of Materials Sciences and Engineering under Award #10122. The PNNL work was performed in the Environmental Molecular Sciences Laboratory, a national scientific user facility sponsored by the Department of Energy's Office of Biological and Environmental Research and located at PNNL.

## Author contributions

Y.L. conceived and supervised the project. H.G. and C.D. carried out the density functional theory calculations under the supervision of M.W.Z. Z.G.Y. and J.N.C. assisted with the data analysis. J.S.S. performed the ellipsometry measurements under the supervision of T.W.N. L.W. and S.A.C. provided the single crystal samples. Y.L. performed the structural characterization and electrical measurements. Y.Z. and M.W. helped with the structural characterization. H.G., C.D., M.W.Z. and Y.L. wrote the manuscript, and all authors commented on the results and the manuscript.

## Competing interests

The authors declare no competing interests.
