## [Peer Review File · Nature Communications]

REVIEWER COMMENTS

Reviewer #1 (Remarks to the Author):

In their manuscript “Ultra-flat and long-lived plasmons in a strongly correlated oxide”, H. Gao, et al. present mostly theoretical and some experimental results on the plasmonic and electronic behavior of the correlated insulator α -TiO₃. Using both experimental ellipsometry, and DFT calculations, the authors demonstrate that the α -TiO₃ permittivity is characterized by in-plane isotropy and out-of-plane anisotropy. This gives rise to two energy ranges over which type-I hyperbolicity can take place (i.e., where the in-plane and out-of-plane permittivities have opposite sign). Theoretical calculations of the loss function show that a flat plasmon band exists within one of these hyperbolic regimes, possessing a higher-than-usual plasmon intensity over a broad range of momenta (including outside of the first Brillouin zone). This is attributed to a vanishing value of $\text{Im}(\epsilon)$ over a wide range of momenta for both in-plane and out-of-plane plasmons. Calculation of the electronic band structure reveals that the long-lived flat-band plasmons originate from transitions between a single flat occupied band and multiple unoccupied flat bands near the Fermi Energy. The authors note that the flat plasmon modes propagate over a much larger range of momenta compared to any other material previously reported.

The authors present a compelling case for the future experimental study of plasmons in α -TiO₃ and offer general criteria for realizing long lived ultra-flat plasmons that may permit realization of this behavior in materials beyond α -TiO₃. It should be noted that no experimental data is presented demonstrating the existence of these plasmons. While I find the overall result interesting and potentially significant to a broad audience, there are some substantial issues that need to be addressed before I can recommend publication in Nature Communications.

Significant Comments:

1. The main conclusion of the paper hinges on the presence of “long-lived” flat plasmons (i.e. a small value of $\text{Im}(\epsilon)$ over a broad range of momenta). I would be more convinced that this is an experimental reality if the authors included $\text{Im}(\epsilon)$ in Fig. 2b from the experimentally extracted ellipsometry data, showing that it is small where $\text{Re}(\epsilon)$ changes sign.
2. The authors offer that the origin of the long lived flat plasmons is in the presence of a single flat occupied band and three unoccupied flat bands near the Fermi Energy. Is it true that none of the materials shown in Figure 5 have this characteristic? If so, this would greatly bolster the author’s main claim and should be stated somewhere in the paragraph discussing Figure 5 (lines 293-300)
3. Related to comment 2: would it be enough to realize this behavior if there were a single flat occupied band and only two unoccupied flat bands? How are flat plasmons influenced by adding/subtracting more flat bands around the Fermi energy? It would be helpful if the authors

could comment on this to see how much the plasmon behavior can be “tuned” around the specific case of α -TiO₂.

4. The text would benefit from a concise statement on why the parity of the wavefunctions residing in the flat bands near the Fermi energy causes the plasmons to be long-lived (i.e., $\text{Im}(\epsilon)$ to vanish). The abstract hints that this is the case, but I am unable to find such a statement in the main manuscript.

5. On line 256, it is stated that LFE is neglected in the calculation of the permittivity, yet, Fig. 3e,f shows that this has a significant effect on the theoretical plasmon behavior (including the energy and number of plasmon branches). The authors should comment on why it is okay to neglect LFE in this situation when it was apparently non-negligible in the case of out-of-plane plasmons in TiO₂.

Additional Comments:

6. The authors should clarify in the caption if Figs. 3a,b includes LFE. It seems like the answer is “yes”, but the caption should indicate this.

7. The clarity of presentation would be significantly improved if the authors kept a consistent color/style convention between Figures 2 and 3. For Figure 2, solid indicates $\text{Re}(\epsilon)$ while dashed indicates $\text{Im}(\epsilon)$, and red indicates out-of-plane while blue indicates in-plane. For Figure 3, red means $\text{Re}(\epsilon)$ while blue means $\text{Im}(\epsilon)$, and solid indicates with LFE while dashed indicates without LFE. This is quite confusing at first, so I would suggest the authors either change the convention in Fig. 2 or change the convention in Fig. 3 so that the two are consistent with one another.

8. Line 128 says “The characteristics of these two ρ vs. T curves attest to the correlated insulating behavior of α -TiO₂.”. The authors should provide a citation or further explanation for why the curve attests to the correlated insulating behavior.

9. Figure 4b and c are labelled incorrectly in the figure + caption, or in the main text (the text refers to Figure 4b when referencing Figure 4c, and vice versa)

10. There are several cases in the text where the authors make a general statement about a result or phenomenon where specific examples are lacking. I would suggest adding specific examples (not just citations) for the following passages:

a. Line 71 refers to “unprecedented functionalities”

b. Line 82 refers to “great potential for practical applications”

c. Line 91-92 refers to “fascinating physical properties and applications”

d. Line 175: “interesting plasmon properties and functionalities”

Reviewer #2 (Remarks to the Author):

Dear Editor

The authors study the plasmon in a strongly correlated oxide α -Ti₂O₃ which has narrow bands due to a strong Hubbard U. They synthesized the single crystals of α -Ti₂O₃ and measured the complex permittivity ϵ by ellipsometry. Their theoretical results of ϵ are overall consistent with the experimental data of the. A branch of dispersionless undamped plasmon band around 2.6 eV has been proposed by studying the electron energy loss spectra using random phase approximation. It is interesting that this flat plasmon band goes over the entire Brillouin zone and in both x(y) and z directions. After examining the electronic transition properties, the authors conclude that the flat plasmon band is mainly attributed to the narrow bands above Fermi energy.

I think this manuscript contains some interesting results and sheds some insight into the understanding/searching unconventional plasmon excitations in three dimension material. In addition, the manuscript is written in a clear manner. However, the method for plasmon dispersions is rather standard and the calculations based on the first-principles methods are straightforward. The flat plasmon band due to narrow bands is also not super surprising, e.g., see [PRB 104, 045140 (2021)] and [arXiv:2011.02982]. Also, the authors did not provide direct experimental data of plasmon and therefore did not establish a very direct and unambiguous connection between their theory and the experimental observations. In view of these considerations, I cannot recommend the current manuscript a publication in Nature Communications but a more specialized journal after the authors consider the points that are listed below.

1. When discussing the origin for their ultra-flat behavior, the author made an approximation by taking long-wavelength limit (Line 261). However, the flat plasmon goes over the entire Brillouin zone but not only around $\mathbf{q}=0$. Maybe the authors should have more comments about the validity of this approximation.
2. Finite temperature is another source of damping of plasmons. I think that the main theoretical results are for zero temperature. How are the finite temperature effects on the lifetime of plasmons?
3. Line 286, the authors said that "Obviously, band 2 consists largely of Ti 3d_{z²} orbital character and has an odd parity". However, it is not obvious to me that the left wavefunction in Fig. 4(b) breaks the inversion symmetry. Maybe the authors can emphasize this point in caption or plot. And maybe the authors should mention that there exist bonding and antibonding orbital. Otherwise, the casual reader may confuse why a 3d_{z²} orbital can have odd parity.

4. Some minor issues should be noticed. For instance, Line 136, the indices α and β are introduced without definition. Line 243, it should be Fig. 4c. Line 258, Eq. (2) has a wrong symbol. And so on.

Reviewer #3 (Remarks to the Author):

The authors present a detailed study of strongly correlated Mott insulator Ti_2O_3 and show that, in bulk single crystal, it has a near dispersionless frequency-wavevector relation. Noteworthy is that they also investigate the physical origins of these flat-dispersion plasmons in this oxide.

The quality of the data seems excellent, and the discussion of their results is well interpreted and sound, with references that are well justified.

These results are potentially significant and will surely help move forward the field of “plasmonic in strongly correlated systems.”

Few additional comments:

The authors should clearly distinguish and compare this work to previous work in similar Ti_2O_3 system, especially include a discussion of their own work in this field (e.g., when the PI was first author...)

A more detailed discussion of plasmon lifetime is necessary since the plasmons discussed here are not affected by Landau damping over a wide frequency range.

Figure 5 should be edited for clarity. Perhaps, increasing the height of this graph would help the reader...

We thank the reviewers for their thoughtful and constructive comments. Below are our point-to-point responses to the reviewers' comments. We carried out new experiments and theoretical calculations, while the complementary results were added as Figure 2b-d, Figure 4b, Supplementary Figures 4-7, and Supplementary Notes 1, 2. Following the suggestions, we have revised our manuscript carefully. We believe that the quality of the revised manuscript has been greatly enhanced, and now we are resubmitting it for your kind reconsideration for publication.

Thank you again for your consideration. We are looking forward to your favourable decision.

COMMENTS TO AUTHOR:

Reviewer #1: In their manuscript "Ultra-flat and long-lived plasmons in a strongly correlated oxide", H. Gao, et al. present mostly theoretical and some experimental results on the plasmonic and electronic behavior of the correlated insulator α -Ti₂O₃. Using both experimental ellipsometry, and DFT calculations, the authors demonstrate that the α -Ti₂O₃ permittivity is characterized by in-plane isotropy and out-of-plane anisotropy. This gives rise to two energy ranges over which type-I hyperbolicity can take place (i.e., where the in-plane and out-of-plane permittivities have opposite sign). Theoretical calculations of the loss function show that a flat plasmon band exists within one of these hyperbolic regimes, possessing a higher-than-usual plasmon intensity over a broad range of momenta (including outside of the first Brillouin zone). This is attributed to a vanishing value of $\text{Im}(\epsilon)$ over a wide range of momenta for both in-plane and out-of-plane plasmons. Calculation of the electronic band structure reveals that the long-lived flat-band plasmons originate from transitions between a single flat occupied band and multiple unoccupied flat bands near the Fermi Energy. The authors note that the flat plasmon modes propagate over a much larger range of momenta compared to any other material previously reported.

The authors present a compelling case for the future experimental study of plasmons in α -Ti₂O₃ and offer general criteria for realizing long lived ultra-flat

plasmons that may permit realization of this behavior in materials beyond α -Ti₂O₃. It should be noted that no experimental data is presented demonstrating the existence of these plasmons. While I find the overall result interesting and potentially significant to a broad audience, there are some substantial issues that need to be addressed before I can recommend publication in Nature Communications.

Authors' Response:

We greatly acknowledge the reviewer for the positive opinion. We have answered all questions from the reviewer below.

[1] The main conclusion of the paper hinges on the presence of “long-lived” flat plasmons (i.e. a small value of $\text{Im}(\epsilon)$ over a broad range of momenta). I would be more convinced that this is an experimental reality if the authors included $\text{Im}(\epsilon)$ in Fig. 2b from the experimentally extracted ellipsometry data, showing that it is small where $\text{Re}(\epsilon)$ changes sign.

Response: We thank the reviewer very much for this important comment. Following the suggestion, we added the $\text{Im}(\epsilon)$ for α -Ti₂O₃ which is obtained from the experimentally extracted ellipsometry data (**R1**). As expected, the $\text{Im}(\epsilon)$ value is small where $\text{Re}(\epsilon)$ changes sign, consistent with our theoretical results (**R3b**). The new experimental results were added to the revised manuscript as new **Fig. 2b**, and corresponding statement was also added, which is highlighted in page 7, lines 16-18.

R1. Real and imaginary parts of the permittivity along the xy -plane and the z - direction, collected by ellipsometry at room temperature. Added as new **Fig. 2b**.

[2] The authors offer that the origin of the long lived flat plasmons is in the presence of a single flat occupied band and three unoccupied flat bands near the Fermi Energy. Is it true that none of the materials shown in Figure 5 have this characteristic? If so, this would greatly bolster the author’s main claim and should be stated somewhere in the paragraph discussing Figure 5 (lines 293-300).

Response: We appreciate the reviewer very much for this constructive comment. As the reviewer mentioned, the long-lived flat plasmons in α -Ti₂O₃ (Mott insulator) originate from the interband transitions between a single relatively flat occupied band and flat unoccupied bands that are renormalized by the strong electronic correlation effect. For the other materials considered in **Fig. 5**, they, except VSe₂, are not strongly correlated systems, thus the renormalized flat energy bands do not exist in these systems.

R2. Band structure and projected density of states of VSe₂. The zero of the vertical axis corresponds to the Fermi level. (Fig. 4a from **P. Cudazzo, et al. Phys. Rev. B 96, 125131 (2017).**)

VSe₂ is a $3d^1$ electron system, same as α -Ti₂O₃. Correlation effect should also exist among those V $3d^1$ electrons, thus renormalized energy bands can be expected in VSe₂. As shown in **R2**, there are indeed some relatively flat unoccupied V $3d$ bands near Fermi level, which is marked with a red rectangle. However, the correlation effect in VSe₂ is not so strong to open a band gap at Fermi level, which makes it a metallic system (**R2**). Its flat plasmon originates from the **intraband** transition with the screening effect of the interband transitions (**P. Cudazzo, et al. Phys. Rev. B 96, 125131 (2017).** **da Jornada FH, et al. Nat. Commun. 11, 1013 (2020).**), which is similar to those in the other non-correlated systems but different from that in α -Ti₂O₃. As we described in the manuscript, α -Ti₂O₃ is a strongly correlated insulator, and its long-

lived flat plasmons originate from the **interband** transitions between the occupied and unoccupied flat bands. Hence, our main claim is further bolstered by this more detailed analysis. We thank the reviewer again for this important comment. **Some clarification has been added to the revised manuscript, which is highlighted in page 15, lines 4-15.**

[3] Related to comment 2: would it be enough to realize this behavior if there were a single flat occupied band and only two unoccupied flat bands? How are flat plasmons influenced by adding/subtracting more flat bands around the Fermi energy? It would be helpful if the authors could comment on this to see how much the plasmon behavior can be “tuned” around the specific case of α -Ti₂O₃.

Response: We thank the reviewer very much for this thoughtful comment. As mentioned in comment 2, the origin of the long-lived flat plasmons in α -Ti₂O₃ is in the presence of a single flat occupied band (α) and three unoccupied flat bands (β , γ , and ξ) near Fermi energy (**R3a**). Combining the analysis of the transition dipole moment matrix, we concluded that the $\alpha \rightarrow \beta$ transition is the main absorption at the energy range of 0 – 3 eV, consistent with the peak at ~ 1.2 eV in the $\text{Im}(\epsilon)$ spectra (**R3b**), and it dominates the emergence of the plasmons in α -Ti₂O₃. According to the theoretical analysis of the dielectric function $\epsilon_{00}(\mathbf{q}, \omega)$, we concluded that one flat unoccupied band (β band) can lead to the dispersionless propagation behavior of the plasmons (in α -Ti₂O₃). Hence, we believe that it would be enough to realize similar behavior if there were a single flat occupied band and two unoccupied flat bands that contributes to the plasmon.

R3. a, The density of states (DOS) (blue solid line) and k ratio (black dotted line) as a function of energy. **(Fig. 4c)** **b**, Real and imaginary parts of the permittivity of α -Ti₂O₃ obtained from first-principles calculations. **(Fig. 2c)**

As mentioned above, flat plasmons can be achieved at least with one occupied flat band and one unoccupied flat band. When adding or subtracting more flat bands from these participating bands, the process of electron transitions (absorption) would be affected. **1.** If the relative flat occupied (unoccupied) band is subtracted that means no relatively flat occupied (unoccupied) band, the plasmon could disappear, because of the absence of a relatively sharp absorption peak that could lead to the emergence of plasmons (interband). **2.** If adding more flat bands near Fermi level, more absorption can be achieved, which would further change the plasmon energy and intensity. As for tuning the plasmon behaviors in α -Ti₂O₃, we believe chemical doping (i.e. V-doping) could be an efficient way. In strongly correlated systems, chemical doping can tune the strength of correlation effect, which further tunes the band renormalization that leads to varied bandwidth, band position, band gap, and thus absorption (M. Uchida, *et al. Phys. Rev. Lett.* **101**, 066406 (2008)). Since the flat plasmons in α -Ti₂O₃ originate from the interband transitions between those correlation-effect renormalized occupied and unoccupied flat bands, their behaviors (including plasmon energy, flatness and intensity) could be tuned by chemical doping. Some clarification has been added to the revised manuscript, which is highlighted in page 15, lines 16-22 and page 16, lines 1-4.

[4] The text would benefit from a concise statement on why the parity of the wavefunctions residing in the flat bands near the Fermi energy causes the plasmons to be long-lived (i.e., $\text{Im}(\epsilon)$ to vanish). The abstract hints that this is the case, but I am unable to find such a statement in the main manuscript.

Response: We thank the reviewer for this helpful comment. According to the optical selection rule, only the interband transitions between two different states with opposite parities are allowed, and thus could have contributions to the $\text{Im } \epsilon$. In this work, we attributed the generation of ultra-flat plasmons (at ~ 2.6 eV) in α -Ti₂O₃ to the absorption peak that is centered at ~ 1.2 eV in the $\text{Im}(\epsilon)$ spectra (R3b) and comes from the transitions between state 2 and states 3/4 with opposite parities (R4a). The Landau damping of these plasmons should be arisen from the interband transitions with the transition energy close to 2.6 eV. As shown in R4b, only the transition between state 1 and state 12 has the corresponding energy difference. However, the wave functions of state 1 and state 12 both exhibit an even parity, so this transition is forbidden (T1). Hence, we get vanished values for $\text{Im}(\epsilon)$ at the plasmon energy, which preserves the plasmons are long-lived. Corresponding statement has been added to the revised manuscript, which is highlighted in page 11, lines 12-19.

R4. a, The Bloch electron wave functions corresponding to the band 1, band 2, band 3 and band 4 at the Γ point. O point is the center of inversion symmetry. “+” and “-” represent the sign of the real part of the wave functions. **(Added as new Fig. 4b.)** **b**, Orbital-resolved band structure of α -Ti₂O₃.

Band index	Reps	Parity	Transition (from 1)	Transition (from 2)
1	A _{1g}	+	--	--
2	A _{1u}	-	--	--
3, 4	E _g	+	×	✓(main)
5, 6	E _g	+	×	✓
7, 8	E _u	-	✓	×
9	A _{2u}	-	✓	×
10, 11	E _u	-	✓	×
12	A _{2g}	+	×	✓

T1. The electronic states at Γ , and their corresponding irreducible representations of D_{3d} , parity of space inversion. **(Table 1)**

[5] On line 256, it is stated that LFE is neglected in the calculation of the permittivity, yet, Fig. 3e,f shows that this has a significant effect on the theoretical plasmon behavior (including the energy and number of plasmon branches). The authors should comment on why it is okay to neglect LFE in this situation when it was apparently non-negligible in the case of out-of-plane plasmons in Ti₂O₃.

Response: We thank the reviewer for this important comment. As the review mentioned, we should consider LFE when we calculate the permittivity. Thus, we recalculated the dielectric function, and the new results are shown in **R3b**. After considering LFE, the first hyperbolic window is nearly unchanged, but the second hyperbolic window shrinks because of the red-shifting of $\text{Re}\epsilon_z(\omega) = 0$ from 3.22 eV to

2.69 eV. This is more consistent with the experimental results (**R1**), and the LFE shows similar impact on the plasmon behavior along the z -direction (**R5**).

R5. Dielectric function (upper panel) and loss function (lower panel) of α -Ti₂O₃ for **a**, $q = 0.044 \text{ \AA}^{-1}$ and **b**, $q = 0.44 \text{ \AA}^{-1}$ along the Γ - T direction, respectively. The data indicated by the solid (dashed) lines are results with (without) the LFE.

However, we still neglect the LFE in the theoretical analysis for the flatness of the plasmons. The reason is shown as follows. If LFE is included in the analysis, the macroscopic dielectric matrix should be expressed as $\epsilon_M(\mathbf{q}, \omega) = 1/\epsilon_{\mathbf{G}=\mathbf{G}'=0}^{-1}(\mathbf{q}, \omega)$. The LFE is responsible for the off-diagonal terms of $\epsilon_{\mathbf{G},\mathbf{G}'}(\mathbf{q}, \omega)$ that is very difficult to be quantified in the subsequent theoretical derivation. As the review mentioned, LFE shows a significant effect on the plasmon behavior along the out-of-plane direction. However, from $q_{\Gamma-T} = 0.044 \text{ \AA}^{-1}$ to $q_{\Gamma-T} = 0.44 \text{ \AA}^{-1}$ (**R5**), the variation of the plasmon energy without LFE is only 57 meV (3.079 eV - 3.017 eV), which indicates the plasmon mode already exhibits a dispersionless behavior without considering LFE. Hence, when we study the origin of the flatness for the plasmons, we ignored the LFE, since it is very difficult to take into account, and it would have a neglectable influence on the flatness of the plasmons. **Some clarification has been added to the revised manuscript, which is highlighted in page 7, lines 18-22, page 8, lines 1-7 and page 12, lines 17-22.**

Additional Comments:

[6] The authors should clarify in the caption if Figs. 3a,b includes LFE. It seems like the answer is “yes”, but the caption should indicate this.

Response: We thank the reviewer for pointing this out. Yes, the results in **Fig. 3a** and **3b** include LFE. **We added the clarification to the caption of Fig. 3, which is highlighted in page 29.**

[7] The clarity of presentation would be significantly improved if the authors kept a consistent color/style convention between Figures 2 and 3. For Figure 2, solid indicates $\text{Re}(\epsilon)$ while dashed indicates $\text{Im}(\epsilon)$, and red indicates out-of-plane while blue indicates in-plane. For Figure 3, red means $\text{Re}(\epsilon)$ while blue means $\text{Im}(\epsilon)$, and solid indicates with LFE while dashed indicates without LFE. This is quite confusing at first, so I would suggest the authors either change the convention in Fig. 2 or change the convention in Fig. 3 so that the two are consistent with one another.

Response: We thank the reviewer for the helpful suggestion. We have unified the plotting standard for **Fig. 2 (R6)** and **Fig. 3 (R7)** with dark and light red indicating $\text{Re}(\epsilon)$ and $\text{Im}(\epsilon)$ along the in-plane direction, while dark and light blue indicating $\text{Re}(\epsilon)$ and $\text{Im}(\epsilon)$ along the out-of-plane direction, respectively. Solid lines indicate including LFE and dashed lines indicate without LFE. **The new figures were added to the revised manuscript and highlighted in page 28 and 29.**

R6. Dielectric properties of $\alpha\text{-Ti}_2\text{O}_3$ crystal. Added as new Fig. 2.

R7. Plasmon behavior for α -Ti₂O₃. Added as new Fig. 3.

[8] Line 128 says “The characteristics of these two ρ vs. T curves attest to the correlated insulating behavior of α -Ti₂O₃.”. The authors should provide a citation or further explanation for why the curve attests to the correlated insulating behavior.

Response: We thank the reviewer for this comment. As we described in the manuscript, α -Ti₂O₃ has an unpaired $3d^1$ electron on the Ti³⁺ ion, which means there is one Ti $3d$ orbital is half-filled. According to the conventional band theory, it should be a metal whose resistivity should be decreased with decreasing temperature. However, experimentally its resistivity is increased with decreasing temperature (Fig. 1d), which violates the conventional band theory and demonstrates that α -Ti₂O₃ is a strongly correlated insulator with the correlation effect between the $3d^1$ electrons should be considered (M. Uchida, *et al. Phys. Rev. Lett.* **101**, 066406 (2008). C. Chang, *et al. Phys. Rev. X* **8**, 021004 (2018). F. Morin, *Phys. Rev. Lett.* **3** 34(1959).). Corresponding clarification and references were added to the revised manuscript, which is highlighted in page 6, lines 5,6.

[9] Figure 4b and c are labelled incorrectly in the figure + caption, or in the main text (the text refers to Figure 4b when referencing Figure 4c, and vice versa)

Response: We thank the reviewer for pointing this out. We made the corrections in the revised manuscript, which is highlighted in page 12, lines 4.

[10] There are several cases in the text where the authors make a general statement about a result or phenomenon where specific examples are lacking. I would suggest adding specific examples (not just citations) for the following passages:

- a. Line 71 refers to “unprecedented functionalities”
- b. Line 82 refers to “great potential for practical applications”
- c. Line 91-92 refers to “fascinating physical properties and applications”
- d. Line 175: “interesting plasmon properties and functionalities”

Response: We thank the reviewer for this comment. We added some specific examples in the corresponding statements:

a. The original sentence is changed to “In strongly correlated electron systems, plasmonic behavior can be drastically altered due to the strong correlation effects, leading to novel properties and unprecedented functionalities. For example, correlation effect with long-range Coulomb interactions could induce unconventional correlated plasmons with multiple plasmon frequencies and low-loss.”, which is highlighted in page 3, lines 11-13.

b. The original sentence is revised to “Interestingly, flat plasmons can transition to localized plasmon wave packets in real-space. By tracking these plasmon wave packets, novel time-resolved plasmonic imaging technique could be realized.”, which is highlighted in page 4, lines 2-4.

c. The original sentence is changed to “As a consequence, α -Ti₂O₃ has a narrow bandgap of ~ 0.1 eV at room temperature which in turn gives rise to fascinating physical properties and applications, such as high-performance mid-infrared photodetection and photothermal conversion”, which is highlighted in page 4, lines 11-14.

d. The original sentence is changed to “Multiple $\text{Re}\epsilon = 0$ points are observed in the experimental and theoretical results, and this unusual predominance making them applicable as functional hyperbolic metamaterials.”, which is highlighted in page 8, lines 9,10.

Reviewer #2: The authors study the plasmon in a strongly correlated oxide which has narrow bands due to a strong Hubbard U . They synthesized the single crystals of and measured the complex permittivity ϵ by ellipsometry. Their theoretical results of ϵ are overall consistent with the experimental data of the. A branch of dispersionless undamped plasmon band around 2.6 eV has been proposed by studying the electron energy loss spectra using random phase approximation. It is interesting that this flat plasmon band goes over the entire Brillouin zone and in both x(y) and z directions. After examining the electronic transition properties, the authors conclude that the flat plasmon band is mainly attributed to the narrow bands above Fermi energy.

I think this manuscript contains some interesting results and sheds some insight into the understanding/searching unconventional plasmon excitations in three dimension material. In addition, the manuscript is written in a clear manner. However, the method for plasmon dispersions is rather standard and the calculations based on the first-principles methods are straightforward. The flat plasmon band due to narrow bands is also not super surprising, e.g., see [PRB 104, 045140 (2021)] and [arXiv:2011.02982]. Also, the authors did not provide direct experimental data of plasmon and therefore did not establish a very direct and unambiguous connection between their theory and the experimental observations. In view of these considerations, I cannot recommend the current manuscript a publication in Nature Communications but more specialized journal after the authors consider the points that are listed below.

Authors' Response:

We appreciate the reviewer for the constructive comments. We have answered all points from the reviewer as follows. **More experimental results were added to the revised manuscript, and the mentioned references were added as ref. 19 and 20.**

[1] When discussing the origin for their ultra-flat behavior, the author made an approximation by taking long-wavelength limit (Line 261). However, the flat plasmon goes over the entire Brillouin zone but not only around. Maybe the authors should have more comments about the validity of this approximation.

Response: We thank the reviewer for this important comment. Yes, we made an approximation when we discuss the origin for the ultra-flat behavior of the plasmons in α -Ti₂O₃ using

$$\varepsilon_{00}(\mathbf{q}, \omega) = 1 - v(\mathbf{q}) \frac{1}{V} \sum_{\mathbf{k}}^{BZ} \sum_{n, n'} \frac{f_{n, \mathbf{k}} - f_{n', \mathbf{k} + \mathbf{q}}}{\hbar\omega + E_{n, \mathbf{k}} - E_{n', \mathbf{k} + \mathbf{q}} + i\eta} \left| \langle \psi_{n, \mathbf{k}} | e^{-i\mathbf{q} \cdot \mathbf{r}} | \psi_{n', \mathbf{k} + \mathbf{q}} \rangle \right|^2 \quad (\text{E1})$$

The approximation $\langle \psi_{n, \mathbf{k}} | e^{-i\mathbf{q} \cdot \mathbf{r}} | \psi_{n', \mathbf{k} + \mathbf{q}} \rangle \approx -i\mathbf{q} \cdot \langle \psi_{n, \mathbf{k}} | \mathbf{r} | \psi_{n', \mathbf{k}} \rangle$ is valid under the long-wavelength limit ($e^{-i\mathbf{q} \cdot \mathbf{r}} \approx 1 - i\mathbf{q} \cdot \mathbf{r}$). Hence, $\left| \langle \psi_{n, \mathbf{k}} | e^{-i\mathbf{q} \cdot \mathbf{r}} | \psi_{n', \mathbf{k} + \mathbf{q}} \rangle \right|^2 \sim q^2$ is valid in the long-wavelength limit. For the three-dimensional system (α -Ti₂O₃), $v(\mathbf{q}) = 4\pi e^2 / \varepsilon_r q^2$.

As a result, the product $v(\mathbf{q}) \left| \langle \psi_{n, \mathbf{k}} | e^{-i\mathbf{q} \cdot \mathbf{r}} | \psi_{n', \mathbf{k} + \mathbf{q}} \rangle \right|^2$ is independent of q in the long-wavelength limit. Then, the flat behavior of the plasmons can be attributed to the relatively flat band $E_{n', \mathbf{k}}$ (conduction band).

Beyond the long-wavelength limit, $e^{-i\mathbf{q} \cdot \mathbf{r}} = 1 - i\mathbf{q} \cdot \mathbf{r} + o(q)$ using the first-order Taylor expansion. Therefore, $\langle \psi_{n, \mathbf{k}} | e^{-i\mathbf{q} \cdot \mathbf{r}} | \psi_{n', \mathbf{k} + \mathbf{q}} \rangle = \langle \psi_{n, \mathbf{k}} | 1 - i\mathbf{q} \cdot \mathbf{r} + o(q) | \psi_{n', \mathbf{k} + \mathbf{q}} \rangle$. Since

$\langle \psi_{n,k} | \psi_{n',k+q} \rangle = 0$, $\langle \psi_{n,k} | e^{-iq \cdot r} | \psi_{n',k+q} \rangle = -i\mathbf{q} \cdot \langle \psi_{n,k} | \mathbf{r} | \psi_{n',k} \rangle + o(q)$. Then, we further studied the product $v(\mathbf{q}) \left| \langle \psi_{n,k} | e^{-iq \cdot r} | \psi_{n',k+q} \rangle \right|^2$ for q extend to $0.5 \times 2\pi / a$ ($2\pi / a \approx 1.14 \text{ \AA}^{-1}$).

According to the Kramers-Kronig relation, the real part of $\varepsilon(\mathbf{q}, \omega)$ can be evaluated from the imaginary part,

$$\text{Re } \varepsilon(\mathbf{q}, \omega) = 1 + \frac{2}{\pi} P \int_0^\infty \frac{\omega' \text{Im } \varepsilon(\mathbf{q}, \omega')}{\omega'^2 - \omega^2} d\omega' . \quad (\text{E2})$$

Therefore,

$$\frac{\partial \text{Re } \varepsilon(\mathbf{q}, \omega)}{\partial q} = \frac{2}{\pi} P \int_0^\infty \frac{\omega'}{\omega'^2 - \omega^2} \frac{\partial \text{Im } \varepsilon(\mathbf{q}, \omega')}{\partial q} d\omega' . \quad (\text{E3})$$

The plasmon mode is determined by $\text{Re } \varepsilon(\mathbf{q}, \omega) = 0$. The flat plasmon means that the plasmon dispersion is independent of q . Thus, it should satisfy $\partial \text{Re } \varepsilon(\mathbf{q}, \omega) / \partial q = 0$, for $\hbar\omega$ at the plasmon energy. That is, the flat plasmon requires that $\partial \text{Im } \varepsilon(\mathbf{q}, \omega) / \partial q = 0$. Notably, the Eq. (E3) is mostly contributed by the term for $\omega' \approx \omega$, i.e., for $\hbar\omega'$ near the energy of plasmons.

Using Eq. (E1), we can obtain the interband part of $\text{Im } \varepsilon(\mathbf{q}, \omega)$:

$$\text{Im } \varepsilon^{\text{inter}}(\mathbf{q}, \omega) = \frac{g}{(2\pi)^3} \pi v(\mathbf{q}) \sum_{v,c} \int_{\text{BZ}} d^3\mathbf{k} \delta(\hbar\omega + E_{v,\mathbf{k}} - E_{c,\mathbf{k}+\mathbf{q}}) F_{v,c}(\mathbf{k}, \mathbf{q}), \quad (\text{E4})$$

where v, c are the band indexes of valence and conductive bands, respectively.

$F_{v,c}(\mathbf{k}, \mathbf{q}) = \left| \langle \psi_{v,\mathbf{k}} | e^{-iq \cdot r} | \psi_{c,\mathbf{k}+\mathbf{q}} \rangle \right|^2$ is the overlap of states. $\delta(x)$ is the Dirac function.

Eq. (E4) can be simplified in the form:

$$\text{Im } \varepsilon^{\text{inter}}(\mathbf{q}, \omega) = \pi J_q(\hbar\omega) v(\mathbf{q}) F(\mathbf{q}, \hbar\omega), \quad (\text{E5})$$

where $J_q(\hbar\omega)$ is the joint density of states (JDOS) for finite wave vector \mathbf{q} , which

is defined as

$$J_q(\hbar\omega) = \frac{g}{(2\pi)^3} \sum_{c,v} \int_{BZ} d^3\mathbf{k} \delta(E_{c,\mathbf{k}+q} - E_{v,\mathbf{k}} - \hbar\omega), \quad (\text{E6})$$

$F(\mathbf{q}, \hbar\omega)$ is the average value of $\left| \langle \psi_{n,\mathbf{k}} | e^{-iq \cdot \mathbf{r}} | \psi_{n',\mathbf{k}+q} \rangle \right|^2$ between the states satisfying $E_{n',\mathbf{k}+q} - E_{n,\mathbf{k}} = \hbar\omega$, which is defined as

$$F(\mathbf{q}, \hbar\omega) = \frac{\sum_{v,c} \int_{BZ} d^3\mathbf{k} \delta(\hbar\omega + E_{v,\mathbf{k}} - E_{c,\mathbf{k}+q}) F_{v,c}(\mathbf{k}, \mathbf{q})}{\sum_{v,c} \int_{BZ} d^3\mathbf{k} \delta(\hbar\omega + E_{v,\mathbf{k}} - E_{c,\mathbf{k}+q})}. \quad (\text{E7})$$

According to Eq. (E5), the requirement for the flat behaviors of plasmons $\partial \text{Im} \varepsilon(\mathbf{q}, \omega) / \partial q = 0$ is satisfied when $J_q(\hbar\omega)$ and $v(\mathbf{q})F(\mathbf{q}, \hbar\omega)$ are both independent of q . In **R8**, we plotted the $\text{Im} \varepsilon$, $J_q(\hbar\omega)$ and λ ($\lambda = v(\mathbf{q})F(\mathbf{q}, \hbar\omega)$) as the functions of q , for $\hbar\omega$ at the energy of plasmons. Interestingly, λ is nearly independent of q , for q extend to $0.5 \times 2\pi/a$ ($2\pi/a \approx 1.14 \text{ \AA}^{-1}$). As expected, $J_q(\hbar\omega)$ is nearly q -independent, due to the relative flat conduction bands in $\alpha\text{-Ti}_2\text{O}_3$ ($E_{c,\mathbf{k}+q}$ is nearly q -independent). As a result, $\text{Im} \varepsilon$ is nearly independent of q for $\hbar\omega$ at the plasmon energy that leads to $\partial \text{Im} \varepsilon(\mathbf{q}, \omega) / \partial q = 0$, demonstrating the flat behaviors of plasmons.

Thus, beyond the long-wavelength limit, the approximation $\langle \psi_{n,\mathbf{k}} | e^{-iq \cdot \mathbf{r}} | \psi_{n',\mathbf{k}+q} \rangle \approx -i\mathbf{q} \cdot \langle \psi_{n,\mathbf{k}} | \mathbf{r} | \psi_{n',\mathbf{k}} \rangle$ and q -independent $v(\mathbf{q}) \left| \langle \psi_{n,\mathbf{k}} | e^{-iq \cdot \mathbf{r}} | \psi_{n',\mathbf{k}+q} \rangle \right|^2$ are both invalid. However, we found that the $\lambda = v(\mathbf{q})F(\mathbf{q}, \hbar\omega)$ (where $F(\mathbf{q}, \hbar\omega)$ is the average value of $\left| \langle \psi_{n,\mathbf{k}} | e^{-iq \cdot \mathbf{r}} | \psi_{n',\mathbf{k}+q} \rangle \right|^2$) is independent of q , which can lead to the flat behavior of plasmons with q -independent $J_q(\hbar\omega)$ (flat conduction bands). Hence, **same conclusion**, that is the flat behavior of plasmons can be attributed to the relatively flat conduction band, **can be made at and beyond the long-wavelength limit**. To simply introduce this mechanism, we used the approximation at the long-wavelength limit in the main text, while the scenario beyond the long-wavelength limit is presented in the supplementary materials.

It should be noted that the analysis is performed without considering the local field effect (LFE). In principle, the results in **R8** should be more q -independent with LFE in the real material system. As for why we did not consider the LFE in the analysis, more details can be referred to the response to comment 5 from Reviewer 1. **This discussion**

was added to the supplementary materials as **Supplementary Note 2**, which is mentioned and highlighted in the revised manuscript in page 14, lines 6-8.

R8. The calculated $\text{Im}[\varepsilon]$, J_q , and λ ($v(\mathbf{q})F(\mathbf{q}, \hbar\omega)$) as the functions of q at the plasmon energy along **a**, Γ - S_0 and **b**, Γ - T directions, respectively. **Added as new Supplementary Fig. 7.**

[2] Finite temperature is another source of damping of plasmons. I think that the main theoretical results are for zero temperature. How are the finite temperature effects on the lifetime of plasmons?

Response: We appreciate the reviewer for this important comment. As the reviewer mentioned, the theoretical results presented in the manuscript before are for zero temperature. And yes, we agree with the reviewer that finite temperature can be a source of damping of plasmons. However, in our case, the finite temperature has neglectable impact on the damping of the plasmons (~ 2.6 eV) in α - Ti_2O_3 . The reason is explained as follows.

1. Theoretical calculations

It is known that the damping of plasmons is determined by the imaginary part of dielectric function $\varepsilon_{G'G'}(\mathbf{q}, \omega)$, which can be written as

$$\text{Im} \varepsilon_{G'G'}(\mathbf{q}, \omega) = -\frac{4\pi}{|\mathbf{q} + \mathbf{G}'|^2} \text{Im} \chi_{G',G'}^0(\mathbf{q}, \omega)$$

where $\chi_{G',G'}^0$ is the non-interacting density response function, which is defined by

$$\chi_{G',G'}^0(\mathbf{q}, \omega) = \frac{1}{\Omega} \sum_{\mathbf{k}} \sum_{n,n'}^{\text{BZ}} \frac{f_{n\mathbf{k}} - f_{n'\mathbf{k}+\mathbf{q}}}{\omega + \varepsilon_{n\mathbf{k}} - \varepsilon_{n'\mathbf{k}+\mathbf{q}} + i\eta} \times \langle \psi_{n\mathbf{k}} | e^{-i(\mathbf{q}+\mathbf{G}')\cdot\mathbf{r}} | \psi_{n'\mathbf{k}+\mathbf{q}} \rangle \Omega_{\text{cell}} \\ \times \langle \psi_{n\mathbf{k}} | e^{i(\mathbf{q}+\mathbf{G}')\cdot\mathbf{r}'} | \psi_{n'\mathbf{k}+\mathbf{q}} \rangle \Omega_{\text{cell}}.$$

The finite temperature can affect the imaginary part of dielectric function through the Fermi distribution $f_{nk}(E)$,

$$f_{nk}(E) = \frac{1}{1 + \exp[(E - E_F)/k_B T]}$$

where E_F is the Fermi energy, k_B and T are Boltzmann constant and absolute temperature, respectively. Then, we examined the effect of finite temperature by recalculating the $\text{Im}(\varepsilon)$ and EELS via setting the temperature values from 0 to 400 K. As shown in **R9**, the $\text{Im}(\varepsilon)$ nearly keeps unchanged with temperature increased to 400 K, indicating the damping of plasmons is nearly not influenced by the temperature. Moreover, the difference for plasmon energy, intensity and the maximum propagating wave vector at different temperatures (**R10**) is very small ($\sim 1\%$). Therefore, the finite temperature has negligible effects on the damping of plasmons in α - Ti_2O_3 , based on the calculations.

It should be noted that temperature can affect the plasmon behavior via affecting electron-phonon interactions and thermal excitations of electrons, which would introduce additional dissipation channels for plasmons (**D. Novko. *Nano Lett.* 17, 11, 6991 (2017); A. Iurov, et al. *Phys. Rev. B* 93, 035404 (2016).**). However, these higher-order decay processes can only affect the plasmon modes that have low energy and close to the phonon energy.

R9. Calculated the imaginary part of permittivity for **a**, *xy*-plane and **b**, *z*-direction, respectively. **Added as Supplementary Fig. 4.**

R10. Calculated EELS with LFE for α -Ti₂O₃ along Γ -S₀ and Γ -T directions at **a**, 0 K; **b**, 100 K; **c**, 200 K; **d**, 300 K; and **e**, 400 K. **Added as Supplementary Fig. 5.**

2. Experimental measurements

In fact, the temperature can change the lattice constants and correlation strength in α -Ti₂O₃, which can affect the band structure **near Fermi level** (indicated by the temperature-dependent resistivity result (**Fig. 1d**)). However, band changing **near Fermi level** can only influence the dielectric function at low energy range, which cannot affect the plasmon modes much since the plasmon energy at much higher energy (~ 2.6 eV). As shown in **R11**, the plasmon energy only changed by ~ 63 meV and ~ 50 meV for in-plane and out-of-plane direction, respectively. And the $\text{Im}(\epsilon)$ keeps nearly vanished at the plasmon energy for all temperatures, and does not change much with temperature. Therefore, the damping of plasmons in α -Ti₂O₃ would not be changed much by finite temperature.

R11. Experimental real (upper) and imaginary part (lower) of the permittivity for **a**, *xy*-plane and **b**, *z*-direction, respectively. Added as Supplementary Fig. 6.

Hence, since the plasmon energy in α -Ti₂O₃ is much larger than the $k_B T$ thermal energy and phonon-related coupling energy, the impact of finite temperature on the damping of plasmons can be neglected (D. Novko. *Nano Lett.* **17**, **11**, 6991 (2017); S. Xue, *et al. Phys. Rev. Lett.* **127**, 186802 (2021).). Thus, the long-lived feature of plasmons in α -Ti₂O₃ is robust for finite temperatures. A discussion about the temperature effect was added to the supplementary material as Supplementary Note 1, which is mentioned and highlighted in the revised manuscript in page 11, lines 19-21.

[3] Line 286, the authors said that “Obviously, band 2 consists largely of Ti $3d_{z^2}$ orbital character and has an odd parity”. However, it is not obvious to me that the left wavefunction in Fig. 4(b) breaks the inversion symmetry. Maybe the authors can emphasize this point in caption or plot. And maybe the authors should mention that there exist bonding and antibonding orbital. Otherwise, the casual reader may confuse why a orbital can have odd parity.

Response: We thank the reviewer very much for this important comment. Following the suggestion, we replotted the isosurfaces of the real parts of wavefunctions for state 1, 2, 3, 4 at Γ point (R4a). The space inversion symmetry point of α -Ti₂O₃ is marked by *O* point. The sign of the real parts of wavefunctions was marked, and shown in different colors. Due to the space inversion symmetry, the wavefunctions at Γ point have certain parities with the space inversion operator *P*. As shown in R4a, state 1, 3, 4 have the even parity with respect to *O* point, whereas state 2 has the odd parity. Now, it is more obvious that state 2 breaks the inversion symmetry. According to Goodenough’s model (Inset of Fig. 1d, L.L. Van Zandt, *et al. J. Appl. Phys.* **39**, 594 (1968).), state 1 and 2 are the corresponding a_{1g} bonding orbital, while state 3 and 4 are the corresponding e_g^π bonding orbital. The new plot was added to the revised

manuscript as new **Fig. 4b**. Corresponding clarification was added to the main text, which is highlighted in page 14, lines 9-14.

[4] Some minor issues should be noticed. For instance, Line 136, the indices α and β are introduced without definition. Line 243, it should be Fig. 4c. Line 258, Eq. (2) has a wrong symbol. And so on.

Response: We thank the reviewer for pointing out these issues.

The indices α and β have been defined: “The optical properties of crystalline solids can be described by a complex permittivity $\varepsilon(\omega)_{\alpha\beta} = \text{Re}[\varepsilon(\omega)_{\alpha\beta}] + \text{Im}[\varepsilon(\omega)_{\alpha\beta}]$, where α, β represent the different Cartesian directions.”

The typo about Fig. 4c has been corrected, so does the Eq. (2).

$$\varepsilon_{00}(\mathbf{q}, \omega) = 1 - v(\mathbf{q}) \frac{1}{V} \sum_k^{BZ} \sum_{n,n'} \frac{f_{n,k} - f_{n',k+\mathbf{q}}}{\hbar\omega + E_{n,k} - E_{n',k+\mathbf{q}} + i\eta} |\langle \psi_{n,k} | e^{-i\mathbf{q}\cdot\mathbf{r}} | \psi_{n',k+\mathbf{q}} \rangle|^2$$

We further checked our manuscript carefully, and corresponding corrections have been made and highlighted in page 6, lines 11-13 and page 13, line 1.

Reviewer #3: The authors present a detailed study of strongly correlated Mott insulator Ti_2O_3 and show that, in bulk single crystal, it has a near dispersionless frequency-wavevector relation. Noteworthy is that they also investigate the physical origins of these flat-dispersion plasmons in this oxide. The quality of the data seems excellent, and the discussion of their results is well interpreted and sound, with references that are well justified. These results are potentially significant and will surely help move forward the field of “plasmonic in strongly correlated systems.”

Authors' Response:

We greatly appreciate the reviewer for the positive evaluation. We have gladly answered the questions raised by the reviewer as follows.

Few additional comments:

[1] The authors should clearly distinguish and compare this work to previous work in similar Ti_2O_3 system, especially include a discussion of their own work in this field (e.g., when the PI was first author...)

Response: We thank the reviewer for this constructive comment. Firstly, it should

be noted that the research interest of the previous works about α -Ti₂O₃ has been focused on its unique and broad metal-insulator transition ($400 < T_{MIT} < 550$ K) for a long time (F.J. Morin, *Phys. Rev. Lett.* **3** 34 (1959); M. Uchida, *et al. Phys. Rev. Lett.* **101**, 066406 (2008); C. Chang, *et al. Phys. Rev. X* **8**, 021004 (2018).). In the last few years, we focused on this material system and did several works from the point-of-view of Ti₂O₃ epitaxial films (Y. Li, *et al. Nat. Commun.* **2019**, **10**, 3149; Y. Li, *et al. Adv. Funct. Mater.* **2018**, **28**, 1705657; Y. Li, *et al. NPG Asia Mater.* **2018**, **10**, 522-532.) and particles (Y. Li, *et al. Chem. Mater.* **30**, 4383-4392 (2018); X. Yu*, Y. Li*, *et al. Nat. Commun.* **9**, 4299 (2018); J. Wang*, Y. Li*, *et al, Adv. Mater.* **29**, 1603730 (2017).). The current work is focused on the plasmon property of α -Ti₂O₃ single crystals, which can be clearly distinguished from the previous works including our own ones.

As for our own previous works, we studied the efficient photothermal conversion (J. Wang*, Y. Li*, *et al, Adv. Mater.* **29**, 1603730 (2017).) and mid-infrared photodetection performance (X. Yu*, Y. Li*, *et al. Nat. Commun.* **9**, 4299 (2018).) of the α -Ti₂O₃ nanoparticles, which originate from its correlation-induced narrow bandgap (~ 0.1 eV) at room temperature. In addition, we also fabricated and studied the epitaxial films of different Ti₂O₃ polymorphs including α -Ti₂O₃, o-Ti₂O₃ and γ -Ti₂O₃. Noteworthily, o-Ti₂O₃ and γ -Ti₂O₃ can only be fabricated in the film-form on substrates via the epitaxial stabilization, and they do not exist in the bulk-form in nature. We further studied the correlation-related electronic reconstructions (Y. Li, *et al. Nat. Commun.* **2019**, **10**, 3149.), polymorph-dependent superconductivity (Y. Li, *et al. NPG Asia Mater.* **2018**, **10**, 522-532.) and polymorph-dependent magnetism (Y. Li, *et al. Adv. Funct. Mater.* **2018**, **28**, 1705657.) in Ti₂O₃. Following the suggestion, we added a discussion to the revised manuscript, which is highlighted in page 4, lines 9-16.

[2] A more detailed discussion of plasmon lifetime is necessary since the plasmons discussed here are not affected by Landau damping over a wide frequency range.

Response: We thank the reviewer for this important comment. As shown in **R7**, the plasmon modes along Γ - S_0 and Γ - T directions exhibit undamped features even in the large momentum. Their Landau damping can be identified by the electronic band structure and the parity analysis of the band wavefunctions. The plasmon modes along both directions are located at ~ 2.6 eV, thus the Landau damping of these plasmons should be arisen from the interband transitions with the transition energy close to 2.6 eV. As shown in **R4b**, only the transition between state 1 and state 12 has the

corresponding energy difference. However, the wavefunctions of state 1 and state 12 both exhibit an even parity, so this transition is forbidden (**T1**) by the selection rule. Hence, we get vanished values for $\text{Im}(\epsilon)$ at the plasmon energy (**R3b**), which preserves the plasmons are long-lived. More details can be referred to our response to the 4th comment from Reviewer 1. **Some clarification was added to the revised manuscript, which is highlighted in page 11, lines 12-19.**

[3] Figure 5 should be edited for clarity. Perhaps, increasing the height of this graph would help the reader...

Response: We appreciate the reviewer for this detailed comment. We have further optimized the **Fig. 5** by increasing the height and changing the colors (**R12**).

R12. The superiority of the plasmons in $\alpha\text{-Ti}_2\text{O}_3$. **Added as new Fig. 5.**

REVIEWERS' COMMENTS

Reviewer #1 (Remarks to the Author):

H. Gao, et al. have significantly revised their manuscript “Ultra-flat and long-lived plasmons in a strongly correlated oxide”. The authors have provided detailed explanations to my questions which are now reflected in the revised manuscript. The authors have also clarified certain passages/figures that I previously felt lacked detail or clarity of presentation. Given these substantial changes, I feel that the manuscript has been improved substantially, and will be of significant interest to the field. Therefore, I recommend publication in Nature Communications.

Reviewer #2 (Remarks to the Author):

Dear Editor,

Thanks a lot for sending me back the authors' reply. The authors have addressed all of my comments in their reply and have summarised the new results/calculations in the supplemental materials. In particular, I appreciated their new experimental data on the real and imaginary parts of the permittivity. The quality of the data seems good. Combining the new data as well as more detailed theoretical arguments, I would like to recommend the publication of this manuscript in Nature Communications.

Best wishes.